# Experimental Study on Seismic Performance of Precast Pretensioned Prestressed Concrete Beam-Column Interior Joints Using UHPC for Connection

**DOI:** 10.3390/ma15165791

**Published:** 2022-08-22

**Authors:** Xueyu Xiong, Yifan Xie, Gangfeng Yao, Ju Liu, Laizhang Yan, Liang He

**Affiliations:** 1Department of Structural Engineering, Tongji University, Shanghai 200092, China; xieyf@tongji.edu.cn (Y.X.); liuju@tongji.edu.cn (J.L.); 2Key Laboratory of Advanced Civil Engineering Materials, Tongji University, Shanghai 200092, China; 3School of Civil Engineering, Suzhou University of Science and Technology, Suzhou 215011, China; gfyao@usts.edu.cn; 4China Railway 24th Construction Bureau Anhui Engineering Co., Ltd., Hefei 230011, China; yanlaizhang.24g@crcc.cn; 5China Construction Science & Technology Group East China Co., Ltd., Shanghai 200126, China; heliang_zjkj@163.com

**Keywords:** UHPC, precast pretensioned beam-column interior joint, seismic performance, anchoring length of strands

## Abstract

The traditional connections and reinforcement details of precast RC frames are complex and cause difficulty in construction. Ultra-high-performance concrete (UHPC) exhibits outstanding compressive strength and bond strength with rebars and strands; thus, the usage of UHPC in the joint core area will reduce the amount of transverse reinforcement and shorten the anchoring length of beam rebars as well as strands significantly. Moreover, the lap splice connections of precast columns can be placed in the UHPC joint zone and the construction process will be simplified. This paper presented a novel joint consisting of a precast pretensioned prestressed concrete beam, an ordinary precast reinforced concrete (RC) column, and a UHPC joint zone. To study the seismic performance of the proposed joints, six novel interior joints and one monolithic RC joint were tested under low-cyclic loads. Variables such as the axial force, the compressive strength of UHPC, the stirrup ratio were considered in the tests. The test results indicate that the proposed joints exhibit comparable seismic performance of the monolithic RC joint. An anchorage length of 40 times the strands-diameter and a lap splice length of 16 times the rebar-diameter are adequate for prestressed strands and precast column rebars, respectively. A minimum column depth is suggested as 13 times the diameter of the beam-top continuous rebars passing through the joint. In addition, a nine-time rebar diameter is sufficient for the anchorage of beam bottom rebars. The shear strength of UHPC in the joint core area is suggested as 0.8 times the square root of the UHPC compressive strength.

## 1. Introduction

Precast RC structures have drawn great attention for their strong points such as fast construction, high quality, and low environmental impact [1]. Connections between precast components are crucial for the integrity of precast structures under seismic loads [2]. Grout sleeves are the most common choice for longitudinal reinforcement connection of precast columns for frame structures [3]. However, the method requires high accuracy to assemble the precast components successfully and there is no effective way to check the grouting quality. In addition, beam-column joints are the vital part of the precast earthquake-resistant frames [4,5]. To meet the requirement of anchorage demands of reinforcements of precast beams and the shear strength of joint zone under seismic actions, the joint reinforcement details of traditional precast frame structures are often complex [6,7]. Especially in high earthquake intensity areas, numerous transverse as well as longitudinal reinforcements are required in the joint zone, which will bring great construction difficulty and deficiency [8].

In order to improve the strength of beam-column joints, FRCC and FRP rehabilitation methods are used and studied. Vecchio et al. [9] proposed a beam-column joint with a FRCC jacketing in the joint zone area. The seismic performance of the proposed joints was studied by cyclic loads. The shear strength and energy dissipation were promoted effectively through the thin jacketing. CFRP sheets and CFRP ropes were used to strengthen the joint strength in the literature [10]. Similar studies and conclusions were carried out by various retrofit methods, including additional bars, steel plates, and angles [11,12,13,14]. It was inspiring to see that such retrofit methods could improve the seismic performance of those old-style RC beam column joints effectively. However, for the precast RC structures which were designed with modern codes, it could be troublesome to use the retrofit methods rather than some new construction material to improve the seismic performance from the very beginning.

Pretensioned prestressed beams have smaller section size than non-prestressed beams. With the use of pretensioned strands, precast pretensioned prestressed beams can carry the live loads in construction stage so that the bracing needed for precast beam construction can be omitted, leading to a lower cost economically [15,16]. However, the development length of pretensioned strands in the joint zone is too long to satisfy. Taking C50 concrete as an example, the anchorage length of typical pretensioned strands is 119 times of the diameter of the strands, that is, 1.8 m for strands with a diameter of 15.2 mm [17].

Ultra-high-performance concrete (UHPC) is an efficient material with the advantages of ultra-high strength and excellent bond strength [18]. With the use of UHPC in the joint zone, the connection details of precast beam-column joints can be simplified and improved [19,20]. The amounts of transverse reinforcements in the joint zone can be greatly reduced due to the high strength of UHPC. Moreover, the high bond strength between UHPC and rebars as well as strands can greatly shorten the anchorage length and lap-spliced length of reinforcement [21,22].

Extensive investigations have been carried out on the bond strength of UHPC and rebars [23,24,25,26,27], mechanical or seismic behavior of precast beams and columns using UHPC for connections. According to the test results, the bond strength between UHPC and rebars was more than 20 MPa and the anchoring length of the straight steel reinforcing bar was 8 to 12 times the rebar diameter [28,29]. Meanwhile, the bond strength was 7 to 12 MPa for prestress strands with diameters ranging from 12.7 mm to 21.8 mm. The development length of strands in UHPC have been suggested as 40 or 42 times the strands diameter by different researchers [30,31]. Maya [32] and Graybeal [33] carried out two full-scale girder tests using lap-spliced strands embedded in UHPC for connection. Approximately 90% of the design ultimate flexural capacity were reached with a strand splice length of 50 times the strands diameter. Peng et al. [34] carried out cyclic-loading tests on six precast columns with lap-spliced connection using UHPC. The lap length of the rebars varied from 10 to 30 times the rebar diameter. The test results showed that with a lap length of 10 times the rebar length, the precast columns were able to achieve the comparable seismic performance of monolithic ones in terms of bearing strength and ductility. Wang et al. [35] conducted cyclic tests on a precast bridge column with lap-spliced connection using UHPC and it was concluded that a lap length of 10 times the rebar diameter was enough for rebars with a diameter of 32 mm to exhibit similar seismic performance of the corresponding monolithic RC column.

On the other hand, few investigations have been focused on the seismic performance of beam-column joint using UHPC in the joint zone. Wang et al. [36] carried out cyclic tests on five exterior and four interior monolithic UHPC beam-column joints and the shear strength of the joint zone were studied. The parameters studied were the axial force, stirrup ratio, and joint type. The test results showed that UHPC specimens exhibited excellent seismic performance. However, all the joints were monolithic UHPC beam-column joints and the rebars were passing through the joint zone continuously. The sufficient anchorage length of a rebar in the UHPC joint zone could not be tested by their experiments. Zhang et al. [37] conducted cyclic tests on four interior precast UHPC/RC composite beam-column joints. The anchorage method of beam bottom longitudinal rebars and the stirrup ratio of the joint zone were set as variables to study the seismic performance of the UHPC joints. The anchorage length of straight or headed bars of beams were suggested as 16 times or 8.1 times the rebar dimeter, respectively. Yet, the connection of column rebars was not focused on in Zhang’s study. The column rebars were passing through the joint zone continuously so that the lap-spliced connection using UHPC was not validated in the paper. Ma et al. [38] tested five precast beam-column connections with lap-spliced bars in UHPC. The column rebars were lap-spliced in UHPC joints and the lap length was 15 times the rebar diameter. The beam rebars were anchored in UHPC joints and the anchorage length was 10 times the rebar diameter with a tail extension length of 5 times the rebar diameter. The shear strength of UHPC joints were investigated. Nevertheless, the precast beams were non-prestressed specimens only. It was widely known that the cross-sections of prestressed beams and ordinary beams were different. In general, the ratio of beam depth to beam width of prestressed beams was larger than that of ordinary beams, which might cause different stiffness ratios of beam to column. In addition, the damping ratio of prestressed frames were smaller than that of ordinary concrete frames, leading to a larger earthquake action for prestressed members. Xue et al. [39] carried out the cyclic tests of two beam-column interior joints and two exterior joints to study the seismic performance of precast joints using UHPC-based connections. The UHPC was used for the connections of beam and column rebars. The anchorage and lap-spliced length of beam and column rebars were both 15 times that of the rebar diameters. The cast-in-place concrete included UHPC part and ordinary concrete part. The seismic performance of precast joints was compared to that of monolithic joints under high axial forces. The test results showed that precast members reached similar bearing strength to monolithic ones, indicating that UHPC-based connections could take full use of the strength of beam and column rebars. Yet, only one interior and one exterior beam-column joint were tested, respectively. The limited number of specimens deterred further investigations.

After the review of studies mentioned above, the remaining problems are as follows. The anchorage length of strands in UHPC can be shortened; so, theoretically, it is feasible to use UHPC to connect the pretensioned beams and precast columns in the joint zone and, herein, the advantages of prestressed members and precast systems are combined together. However, the investigation on the application of UHPC in the joint zone for precast prestressed beam-column joints is not found by the authors. There are significant differences between prestressed members and ordinary members in section size ratios. In addition, the contribution of pretensioned prestressed strands at the beam end in seismic actions was unclear. Herein, a new type of precast frame is proposed in the study. The UHPC is used in the joint zone for connection. The precast beams contain prestressed strands and protruding longitudinal rebars. Both strands and rebars are to be anchored in UHPC. The column longitudinal rebars are lap-spliced in the UHPC joint zone so that the connections of precast beams and columns can be placed within the joint zone by UHPC and the advantages of prestressed members are introduced into the new joints at the same time.

In this study, six precast interior beam-column joints and one monolithic joint were tested on cyclic loads to study the seismic performance. The beams were precast beams with pretensioned strands at the bottom and the columns were precast columns protruding the longitudinal rebars. UHPC were used in the joint zone for fabricating the precast beams and columns together. The effects of axial force, compressive strength of UHPC, stirrup ratios, and bond behavior of rebars in UHPC on the seismic performance of the new joints were thoroughly studied. Moreover, design recommendations are given based on the test results.

## 2. Experimental Program

### 2.1. Specimen Description

Because of the high cost of UHPC, it is economically rational to apply UHPC only in the joint core area, to connect precast pretensioned prestressed beams and precast columns together. The processes of assembly are as follows:Precast pretensioned concrete beam units with protruding bottom rebars and strands are prefabricated and then placed on the precast lower column unit with protruding longitudinal rebars which are to form lap splices later;Transverse reinforcements are set up to the protruding longitudinal rebars of the lower column unit, the position of which are in the joint core area, actually;Precast upper column unit with protruding longitudinal rebars which are to form lap splices together with the rebars of the lower column unit in the joint core area is erected with proper leveling and bracing methods;The continuous top longitudinal rebars are placed in the beam topping and pass through the joint area of beam-column sub-assemblages;UHPC is poured into the joint core area to connect the precast beams and columns together;Precast slabs are placed in site and normal concrete is poured into the beam/slab topping.

The layout of specimens is shown in Figure 1. The section size of columns was 360 × 360 mm. The section size of precast beams was 230 × 350 mm and the height of the beam topping was 100 mm. The length of the columns and beams were 2500 mm and 4700 mm, respectively. The concrete with a class of C50 was used for precast parts and the monolithic RC joint. The reinforcement layouts in the beams were identical to five UHPC precast specimens other than PI2, including four top rebars with diameters of 20 mm. In addition, for specimen PI2, five top rebars with diameters of 18 mm were used to investigate the bond behavior of rebars in UHPC. Two rebars with a diameter of 14 mm and two pretensioned strands with a diameter of 12.7 mm were placed at the bottom of precast beams. The initial stress of strands was set as 0.7 
$f_{\mathrm{pu}}$
. 
$f_{\mathrm{pu}}$
 denotes the ultimate tensile strength of strands. The anchoring length of the beam bottom reinforcing bar and strands were set as 20 times that of the diameter of rebars and 40 times that of the diameter of strands, respectively. The stirrups of the joint zone for UHPC specimens other than PI2 were 10 mm diameter two-leg hoops with 225 mm spacing. As for PI2, the spacing of stirrups in the joint was 90 mm. The layouts of the precast column reinforcement were identical with all specimens, containing eight 25 mm diameter longitudinal rebars and two 16 mm diameter rebars. Specimen PI4 was designed as the standard specimen. Some important factors were considered, as shown in Table 1. The axial force was 600 kN for specimen PI1. The diameter of beam top rebars and the amount of stirrup were increased for specimen PI2. UHPC with a designed compressive strength of 120 MPa was used for PI3 and the volume of steel fibers of UHPC for PI5 was 1% instead of 1.5%. As for PI6, the longitudinal rebars of the column were continuous for PI6. As for the monolithic RC specimen RI1, the beam top and bottom reinforcements were four rebars with 20 mm diameter and four rebars with 18 mm diameter, respectively. The stirrups of the joint were set as four-leg hoops with 12 mm diameter and 90 mm spacing, as shown in Figure 1b.

### 2.2. Specimen Construction

First, the wood formworks were conducted as the reinforcements of precast pretensioned prestressed beams were set, which can be seen in Figure 2a. In order to maintain UHPC and precast specimens integrated, the precast beam-UHPC interface should be designed with specialness. The 30 mm grooves were set as shear keys at the end of precast beams, as shown in Figure 2d. The strands were pretensioned before the concrete of the precast beam was poured and after 7 days, the strands were cut to release. When the concrete strength of precast beams and columns met the requirements, the joints were fabricated together, as seen in Figure 2d. Then, UHPC was poured in the joint zone without vibration due to its excellent fluidity, as can be seen in Figure 2e. The monolithic RC specimens and the precast beam-column connections were placed under same conditions for 28 days.

### 2.3. Material Properties

The UHPC material in the study contains 1% or 1.5% straight steel fibers by volume. The length and diameter of the steel fiber were 13 mm and 0.2 mm, respectively. The tensile strength was 2913 MPa and the elastic modulus 210 GPa. The detailed proportions of the used UHPC are listed in Table 2. The cylinder compressive strength of C50 and UHPC used in the cyclic programs are listed in Table 3. The cylinder compressive strength of C50 was calculated by Equation (1) and that of UHPC was tested according ACI318-14 [40]. The yield as well as ultimate strength of the HRB400 steel reinforcing rebars and prestressed strand are listed in Table 4.

(1)
$$f_{c}^{\prime }=0.8f_{cu}$$

where 
$f_{c}^{\prime }$
 and 
$f_{cu}$
 denote the cylinder and cube compressive strength of concrete.

### 2.4. Test Procedure

The test setup and boundary conditions of the beam-column joint specimens are shown in Figure 3. The base and top of the column were supported by a hinge and two hydraulic actuators were connected to the left and right beam. The positive direction was defined as when the actuators at the beam ends moved towards different directions so that the beam-column connection would clockwise rotate. The distance between the two actuators was 4200 mm. A constant axial force of 200 kN (600 kN for PI2 specimen) was applied to the columns by another actuator.

Figure 4 shows the loading protocol, which is specified by ACI 318-14 and adjusted based on the conditions of test lab and specimen size. Displacement control were applied in the whole test procedure. After applying the axial force, increasing displacements were applied one cycle at each level before the predicted yield displacement (
$\Delta_{y}$
). After that, each displacement was increased by 
$\Delta_{y}$
 and cycled for three times. The test stopped until the failure of specimens.

The loads induced by the three actuators were measured by the computer-controlled ultrasonic pulser and the displacements were measured by LDVTs. Strain gauges were previously set on key positions of the strands as well as longitudinal rebars of beams and the longitudinal rebars of columns and stirrups of joints to record the strains throughout the cyclic programs.

## 3. Results

The whole process of loading could be roughly divided into four stages, that is, (a) cracking stage; (b) yielding stage; (c) peak stage; (d) ultimate stage. The direction is defined as positive for discussion convenience when the two actuators move oppositely causing the joint to rotate clockwise.

(a) Take specimen PI4 for example, the first two flexural fine cracks were found at the right beam end near the column and at the left beam end near the interface of the precast beam key groove and cast-in-place UHPC at the displacement of −2 mm. The first crack showed up in the UHPC joint area at the displacement of +10 mm. The width was less than 0.02 mm. (b) The right and left beam longitudinal rebars yielded at the load of +91.10 kN and +71.35 kN, respectively, in the positive direction according to the equivalent elastoplastic energy method suggested by Park [41]. The corresponding yield displacements were +24.56 mm and +28.97 mm, respectively. At the yield stage, the crack between the precast beam and UHPC joint became quite thick with a width of 1.4 mm and 2.2 mm. The main flexural crack went along the edge of key grooves. The UHPC in the joint area was cured in the natural environment maintenance condition. This may cause large shrinkage of UHPC which led to the relatively weaker performance at the interface. The crack widths in the UHPC joint area were quite small with the maximum value of 0.06 mm, which proved the excellent crack control performance of UHPC. It should be mentioned that the number of cracks at the bottom part of the beam was much smaller than at the top part. The reason could be that the bond strength between concrete and rebars was higher than that between concrete and strands [42]. The tensile stress could be well transferred and cracks were distributed over a longer distance. Diagonal fine cracks with close distance appeared in the UHPC joints area and the maximum crack width was 0.08 mm. (c) As the loads increased, flexural cracks within the plastic hinge zone of beams grew and some of them turned into flexural-shear cracks. The widths of the main cracks at both sides of the interfaces between precast beams and UHPC joint were wider. The maximum loads reached 100.38 kN and 78.86 kN for the right and left beams. The maximum width of cracks in the UHPC joint was 0.1 mm only. (d) As the displacement went up, a typical beam flexural failure occurred. With the cover concrete in the plastic hinge zone spalling and rebars exposing, the loads decreased to 85% of the peak loads and the test was stopped. During the test, the damage of UHPC joint was minor and the design target of strong-joint and weak-members was achieved successfully. The failure mode and crack pattern can be seen in Figure 5d.

For the other five precast UHPC joint specimens, which are exhibited in Figure 5, the test behaviors as well as failure modes were similar compared to PI4. The maximum width of diagonal cracks in the UHPC joint zone was 0.15 mm, indicating the damage of the joint zone was minor and all precast UHPC specimens exhibited the beam flexural failure.

As for the monolithic RC specimen RI1, the crack pattern was different. The distribution of cracks at the beam bottom was broad and the crack number was a lot larger than that in the precast joints as seen in Figure 5f. This is because the bond performance between concrete and rebars is better than strands. Moreover, the crack pattern in the joint zone was different for specimen RI1. When the displacement reached 75 mm, the evident diagonal crack showed up in the joint zone and the maximum crack width was 1.7 mm. As seen in the right bottom of Figure 5f, the failure mode of RI1 exhibited a typical joint shear failure after yielding of the beam longitudinal rebars. Even though the reinforcement details in the joint zone followed the requirements of code standards, the strong members–weak joint philosophy was not fully achieved in the test. Nevertheless, the shear failure occurred at the displacement drift ratio of 3.6%, which already exceeded the required value of 2%. Thus, the seismic performance of specimen RI1 was satisfied in some way. Anyhow, the differences of failure modes between UHPC joint and RC joint showed the advantage of the application of UHPC.

## 4. Analysis and Discussion

### 4.1. Load–Displacement Curves

The hysteretic load–displacement curves of all the specimens are shown in Figure 6. Table 5 lists the yielding load 
$P_{y}$
 and the corresponding displacement 
$\Delta_{y}$
, the peak load 
$P_{m}$
 and the corresponding displacement 
$\Delta_{m}$
, the ultimate load 
$P_{u}$
 and the corresponding displacement 
$\Delta_{u}$
. Figure 6 and Figure 7 show the hysteretic and the envelope load–displacement curves.

For the typical specimen PI4, the hysteretic curves were almost linear before the predicted yield displacement. The load reached the maximum value of +78.86 kN and −101.93 kN for the left beam at the displacement of +56.44 mm (drift ratio = +2.69%) and −50.56 mm (−2.41%). The maximum load of the right beam reached +101.39 kN and −70.25 kN at the at the displacement of +39.74 mm (+1.89%) and −54.88 mm (−2.61%). As the displacement increased to +86.87 mm (+4.14%) in the positive direction for the left beam and −87.58 mm (−4.17%) in the negative position for the right beam, the load maintained about 0.96 times that of the peak load. The strength degradation was quite slight although the displacement was very large. This was caused by the existence of strands at the bottom of the beam. The bond slip was not evident based on observation of the plump hysteretic load–displacement curves of beams. The curves also proved that the strands were able to yield and the anchorage length of strands was adequate.

For the specimen PI1, the axial ratio was raised up from 0.03 to 0.1, the hysteretic curves showed comparable features to those of PI4, as shown in Figure 6a,b. The result indicated that a relatively lower axial ratio had little impact on seismic performance for the joints that suffered from beam flexural failure. However, as the displacement increased to −75 mm for the left beam, the curves became pinching due to the severe concrete spalling of left beam end which can be seen in Figure 5a.

For the specimens PI2 and PI3, the amounts of stirrups of the joint zone increased and the compressive strength of UHPC in the joint zone increased, respectively. The hysteretic curves were almost identical to those of PI4, as shown in Figure 6b,c. As for the specimen PI5, UHPC with a lower compressive strength was used in the joint zone, and the hysteretic load–displacement curves for the left beam were pinching during the large displacement stage, as shown in Figure 6e. This indicated that the strands in the left beam of PI5 might slip due to the insufficient anchorage length. For the specimen PI6, the longitudinal rebars of the column were continuous rather than in lap-spliced connection. The comparison of the hysteretic load–displacement curves of PI4 and PI6 indicated that the seismic performance of the two specimens was quite similar, as shown in Figure 6f, though the peak load of PI6 was slightly higher due to a better integrity.

The hysteretic load–displacement curves of the specimen RI1 can be seen in Figure 6g. The failure mode of RI1 was the joint shear failure followed by beam longitudinal rebars yielding, which is shown in Figure 5g. The typical shear crack caused the joint shear failure after the displacement drift increased to 3.6%. Therefore, the hysteretic load–displacement curves started to pinch at the displacement of 75 mm because of the diagonal shear cracks. It should be mentioned that the reinforcement of the bottom beam for RI1 was 80% of that of the top beam. As a result, the peak load of the left beam in the positive direction and the right beam in the negative direction should be about 80% of that in the other direction. The test data showed about only 10% decrease in the peak load.

Generally, all the six UHPC precast specimens encountered typical beam flexural failure and the use of UHPC in the joint zone changed the failure mode from joint shear failure to beam flexural failure, indicating that the strong-joint and weak-beam philosophy of design was achieved. From the hysteretic curves, the precast specimens showed comparable seismic performance to the monolithic RC joint. With the use of UHPC in the joint area, the stirrup ratio was decreased from 1.74% to 0.24%. Meanwhile, the length of lap-spliced connection with column reinforcement can be shortened to 16 times the diameter of the rebar. As mentioned in the literature [43], ordinary concrete was used in the joint core area of precast beam column joints. As a result, prestressed strands of some specimens were fractured during the tests. However, in the authors’ tests, UHPC provided excellent bond strength in the joint core so that all the strands did not exhibit fracturing in the test procedure. Compared with the load–displacement curves in Maya’s study [18], it was clear that the usage of UHPC in the entire beam-column joint core zones instead of only in the rebar-embedded areas increased the energy dissipations greatly.

The load–displacement envelopes for the seven specimens are shown in Figure 7. For the left beams in the negative direction, PI1-PI5 nearly coincided with each other and the curve of PI6 remained stable for the final stage because of the good integrity. However, in the positive direction, the peak load of PI5 was lower than others. As mentioned previously, the reason could be that for UHPC of which the compressive strength was lower than 120 MPa, the anchorage length of 40 times that of the strand diameter was not enough. In addition, for PI1, the curve dropped in the positive direction, this could be the impact of a higher axial ratio. In addition, even though the monolithic specimen had a higher peak load in the positive direction for left beams, nearly all the precast specimens reached the theoretical bearing strength which was demonstrated in a black horizontal line in the figure, meaning that the strengths of precast specimens met the design standards. Moreover, the areas of the beam bottom reinforcement of precast and monolithic specimens were not exactly the same.

As shown in Table 5, the ductility factor was defined as the ratio of ultimate displacement to yielding displacement. In addition, the values of ductility factor ranged from 2.64 to 4.27 and most of them were larger than 3. It showed that the specimens exhibited satisfying ductility. It should be mentioned that the strands were not as ductile as the rebar and this phenomenon could also be proved by the ductility data.

### 4.2. Energy Dissipation

The equivalent damping coefficient 
$h_{e}$
 [44] is a vital index accounting for the energy dissipation efficiency. In general, the equivalent damping coefficient is defined as follows:
(2)
$$h_{e}=\frac{1}{2\pi }\frac{S_{aecf}}{S_{\Delta oab}+S_{\Delta ocd}}$$

where 
$S_{aecf}$
, 
$S_{\Delta oab}$
 and 
$S_{\Delta ocd}$
 indicate the areas surrounded by the curves shown in Figure 8.

The relationship between the equivalent damping coefficient and the displacement for the seven specimens is shown in Figure 9. For right beams, except specimen PI5, the rest of the precast specimens exhibited higher values of 
$h_{e}$
 than the monolithic RC specimen and the values kept increasing as the drifts were below 4%. After the yielding stage, the 
$h_{e}$
 values range from 0.15 to 0.2 for the precast specimens. When the drift reached 4.3%, the 
$h_{e}$
 decrease resulted from the fact that the cover concrete of precast beams spalled severely. In conclusion, the UHPC precast specimens show a better energy dissipation performance than RI1. In Zhang’s study [37], the 
$h_{e}$
 values reached 0.2 when the drift reached 2%, which showed similar seismic performance to the authors’ specimens.

### 4.3. Degradation of Strength

The strength degradation [45] was defined as when a structural component loses its bearing strength for repeated cycles at the same displacement. The strength degradation coefficient 
$D_{j}$
 was introduced to evaluate the performance of strength degradation. Herein, 
$D_{j,2}$
 and 
$D_{j,3}$
 mean the ratio of the second and third cycle peak load to the first cycle at the same displacement, respectively. The curves of 
$D_{j,2}$
 and 
$D_{j,3}$
 versus displacement of all specimens are shown in Figure 10 and Figure 11.

The values of 
$D_{j,2}$
 was basically above 0.9 and fluctuated in a very small range, indicating that the damage during the second cycle was limited. As for the strength degradation coefficient 
$D_{j,3}$
, the precast UHPC specimens exhibited similar degradation to the monolithic RC specimen in general. The degradation was up to 20% even when the drift increased to 3.8%. Therefore, we could safely draw the conclusion that for precast UHPC specimens who exhibited typical beam flexural failure, the axial ratio, the stirrup ratios within the joint area, and the compressive strength of UHPC as well as the lap-spliced connection for column longitudinal rebars had little influence on the strength degradation.

### 4.4. Degradation of Stiffness

The stiffness degradation was a typical behavior for many structural components subjected to reverse cyclic loading [46]. Figure 12 depicts the curves of secant stiffness of seven specimens versus the displacement. It should be mentioned that for left and right beams, the stiffness in two directions were different because the different reinforcements at top and bottom of the beams. From Figure 12, for the left beams in the negative direction and right beams in the positive direction, the precast specimens exhibited higher stiffness than monolithic ones. Both precast and monolithic specimens followed the similar stiffness degradation rules and UHPC precast specimens performed well in terms of stiffness degradation.

### 4.5. Joint Shear Behavior

The joint shear behavior is vital while earthquakes happen for the joints connecting beams and columns. Here, the shear deformation γ was studied and calculated by Equation (3), where the variables used are defined in Figure 13. Additionally, the shear stress γ in the joint zone is determined by Equation (4), which is recommended by ACI 352-02 code [47].

(3)
$$\gamma =\frac{\sqrt{a^{2}+b^{2}}}{2ab}\left(\left|m_{1}-n_{1}\right|+\left|m_{2}-n_{2}\right|\right)$$


(4)
$$v=\frac{C+T-V_{\mathrm{col}}}{h_{c}b_{j}}$$

where 
C
 is the compressive force at beam end; 
T
 is the tensile force at beam end; 
$V_{\mathrm{col}}$
 is the shear force from the column; 
$h_{c}$
 is the column depth in the direction of loading; 
$b_{j}$
 is the effective width of the joint.

The shear stress versus shear deformation curves of specimen PI1 and RI1 are shown in Figure 14. The maximum shear deformation of the UHPC joint was less than 0.001 rad, which coincided with the results mentioned in the work by Zhang et al. [37]. Nevertheless, the shear deformation of RI1 reached its maximum value of 0.003 rad. The diagonal crack width in the joint area of UHPC specimens was 0.15 mm for maximum value only and for RI1, the value was 1.7 mm. The comparisons show that the damage caused by shear action for UHPC specimens was minor.

The maximum shear stresses of all specimens are summarized in Table 5. From the literature, Wang [36] carried out studies on the shear strength of seventeen monolithic UHPC beam column connections and the specimens were designed by the weak joint–strong specimen principle; therefore, shear failure could occur in the joint zone. Ma [38] studied the shear strength of three precast UHPC interior joints and shear failure happened. Zhang [37] studied the seismic performance of four precast UHPC interior joints and all specimens exhibited beam flexural failure. Based on the literature mentioned above and together with the six specimens in the authors’ test, the normalized shear stresses versus the compressive strength of UHPC scatter data were plotted in Figure 15. The normalized shear stress was defined as the shear stress divided by the square root of the compressive strength of UHPC, 
vfc′
.

In the American structural code (ACI 318-14), the nominal shear stress of 
$\nu_{n}=1.25\sqrt{f_{c}^{\prime }}$
 can be achieved for interior beam-column joints. However, this is only suitable for the normal-strength concrete joint with the joint hoops that meet the requirements of the ACI code. Based on Paulay and Priestley’s [48] theory, the shear strength of a beam-column joint consists of two mechanisms, the strut and truss mechanism. Considering the fact that UHPC has ultra-high compressive strength, it is quite unrealistic and unnecessary for UHPC joints to be equipped with such large amounts of hoops. So, based on the trend in Figure 15, a conservative conclusion is that the nominal shear stress of 
$\nu_{n}=0.8\sqrt{f_{c}^{\prime }}$
 can be defined as the shear strength of UHPC joints without any hoops. In addition, with this formula, the minimum shear strength of the six UHPC specimens was 8.6 MPa and this demonstrates that the stirrups in the joint zone of the six specimens can be removed totally without any safety concerns. This conclusion was consistent with the results of Zhang’s [37] and Ma’s [38] studies.

### 4.6. Strain Analysis

The strains of reinforcements were recorded during the test. The strains of stirrups of UHPC precast specimens were lower than 1500 
με
. The fact that stirrups of UHPC precast specimens did not yield in the test can attribute to the outstanding performance on crack control of UHPC and the ultra-high compressive strength of UHPC. Therefore, for UHPC beam-column joints, the stirrups can be omitted, even the reinforcement ratio of the beam reached a relatively high level (1.21%) based on the test results. On the other hand, the stirrups of monolithic RC specimen yielded with a maximum strain of 2626 
με
. The high stirrup ratio of RI1 (1.74%) was unable to prevent the specimen from suffering a shear failure in the joint.

The bond behavior of rebars in UHPC is a critical issue for seismic performance; so, it needs to be examined with the method reported in [49,50]. As exhibited in Figure 16, the gauges T1, T2 and T3 were set at the UHPC joint zone. Based on the assumption of a bilinear model for the constitutive law of rebars, the stress of the reinforcement at the monitored places were calculated and are displayed in Figure 17. The top continuous rebars through the joint zone were subjected to tension at one face of the joint and compression at the opposite face for the interior joint. As a result, the excessive bond demands and the cyclic loads may cause a bond failure if the anchorage length are not satisfied. The distribution of bond stress along the rebar was not uniform. Taking Figure 17a as an example, in the positive direction, the stresses at T2 and T3 were nearly the same. This is because the anchorage length between T1 and T3 is longer than needed. The maximum stress differences between T1 and T3, 
$\Delta \sigma_{1}$
, as well as those between T1 and T2, 
$\Delta \sigma_{2}$
, are given in Table 6. The distance between T1 and T2, 
s
, and the bond stress of continuous reinforcement in UHPC, 
τ
, are summarized in Table 6. We note that 
τ
 was calculated using 
$\Delta \sigma_{2}$
 and the corresponding data 
s
 for the reason mentioned above. Data for PI6 were untrustworthy because the stress differences between gauges were too big so they were omitted. Moreover, the 
τ
 calculated for PI5 was also puzzling: for the compressive strength of UHPC used in PI5, it was the lowest yet the bond stress was the highest. The data captured by gauges of PI5 may be questionable and the result was going to be abandoned by the authors. The bond stress of specimen PI1 was higher than others because of the higher axial force.

The anchorage length needed for beam top continuous rebars passing through the joint can be calculated by Equation (5):
(5)
$$\pi d_{b}\tau l=\frac{\pi }{4}d_{b}{}^{2}\Delta \sigma$$

where 
$d_{b}$
 is the diameter of the rebar; 
τ
 is the average bond stress between rebars and UHPC; 
l
 is the minimum anchorage length needed; 
Δσ
 is the stress difference of the rebar when it was tensioned on one side and compressed on the other.

The results show that the bond stress can achieve the level of 
$1.2\sqrt{f_{c}^{\prime }}$
 for rebars in the UHPC which almost coincide with the conclusion that is drawn in the literature [51,52,53] that the basic bond strength is 
$1.25\sqrt{f_{c}^{\prime }}$
. A relatively conservative suggestion can be given that for UHPC with the compressive strength higher than 120 MPa, the minimum column depth would be suggested as at least 13 times the diameter of the beam top continuous rebars that passed through the joint zone. This value was 35% less than that required in ACI standards [40], namely 20 times the diameter of beam top rebars. This means that rebars of larger diameters can be used in the moment frames when the UHPC was utilized in the joint zone and this may also decrease the construction difficulty in practice. For beam bottom rebars that were anchored straightly within the UHPC joint area, similar calculations can draw the conclusion that the anchorage length for them is 
$9d_{b}$
. In Xue’s study [39], the anchorage length of beam longitudinal rebars was suggested as 
$15d_{b}$
. The compressive strength of UHPC used in Xue’s tests were a lot higher than that of the authors’ tests. Yet, the suggestions from Xue were conservative according to the authors’ tests. As for the lap-spliced length of precast column reinforcements, it was not a focus of this paper and a conservative suggestion of 
$16d_{b}$
 could be given based on the test results. A further study may be needed for a precise and solid conclusion.

### 4.7. Performance Evaluation

For code ACI 374.1-05 [54], the following acceptance criteria were to be satisfied for beam-column joints of special moment frames at the third cycle hoop at the drift ratio of 3.5%.
The remaining peak load at the limiting drift cycle should not be less than 3/4 of the maximum load of the whole test in the same direction;The energy dissipated at the limiting drift cycle should not be less than 1/8 of the idealized elastoplastic energy of the drift ratio;The residual secant stiffness between ±1/10 of the limiting drift ratio should not be less than 5% of the initial stiffness of the first cycle.

Based on the three acceptance criteria, the results are listed in Table 7. All the specimens meet the criteria, indicating that all of them exhibited satisfying seismic performance under cyclic loads. It should be clarified that for RI1, the shear diagonal cracks developed at the displacement of 90 mm which was at the drift ratio of 4.3%. So, until the drift ratio of 3.5%, the monolithic specimen still remained in satisfying condition. In addition, the residual secant stiffness ratio of precast UHPC specimens were higher than RI1, because the UHPC joint remained intact and the shear deformation was not evident and, at the same time, the UHPC joint provided better bond conditions. In conclusion, the seismic performance of all precast UHPC specimens were comparable and even better than the cast-in-place RC specimen so that the emulative design philosophy can be applied to precast UHPC joints.

## 5. Conclusions

Based on cyclic loading tests of six pretensioned precast UHPC beam-column joint specimens and one monolithic RC specimen, the application of cast-in-place UHPC for connections of precast beams and columns are investigated. The effects of axial force, compressive strength of UHPC, and stirrup ratio in the joint area are investigated. The conclusions can be drawn that:With the general method of curing at room temperature, the compressive strength of UHPC can achieve at least 115 MPa. This makes using UHPC in the joint area within the building construction process for precast structures simple and feasible. The simplified method of precast member connections will speed up the construction of precast systems with great economic benefits.The main cracks develop at the interface of UHPC and precast beams along the 30 mm grooves when the deformation was large. Additional shear-resistance reinforcements can be placed at the interface to improve the integrity performance. Still with the main cracks remaining, the UHPC precast specimens show comparable seismic performance of monolithic RC specimen according to ACI 375-05 acceptance criteria.For UHPC of which the compressive strength is larger than 120 MPa, an anchorage length of 40 times the diameter of the strands in the UHPC joint zone is adequate to develop the design yielding stress under cyclic loads. The short anchorage length of strands in UHPC makes the pretensioned precast frames connected by UHPC in the joint zone practicable. For the beam top continuous rebars passing through the UHPC joint, the minimum depth of columns is suggested as 13 times the diameter of the top continuous rebar. The minimum column depth is reduced by 35% compared to requirements of ACI 318-14. The anchorage length for beam bottom rebars anchored straightly in the UHPC joint zone is suggested as 9 times the diameter of the rebar. With the use of UHPC in the joint zone, the congestion of beam bottom longitudinal rebars can be avoided. The lap-spliced connection for column longitudinal rebars in the UHPC joint zone is reliable, exhibiting an alternative method for the connection of precast column rebars other than the grout sleeves connection. Based on the test results, the lap-spliced length is suggested as 16 times the diameter of the column longitudinal rebars for those of which the diameters are less than 25 mm.In the study, the seismic performance of PI4 with the stirrup ratio of 0.24% and PI2 with the stirrup ratio of 0.61% was quite similar. The shear stress of the UHPC specimen reached 0.6
$\sqrt{f_{c}^{\prime }}$
 and the joint remained intact and the maximum width of diagonal cracks of UHPC joints was 0.15 mm. The investigation based on the test results from the literature shows that the shear strength of UHPC joint can be suggested as 0.8
$\sqrt{f_{c}^{\prime }}$
. The stirrups of the UHPC joints could be eliminated totally with safety when the shear stress is less than the suggested shear strength.The precast pretensioned concrete beam-column joint using UHPC for connection proposed by the authors has the advantage of simple connection for precast beams and columns and comparable seismic performance to a monolithic RC joint. It provides a new way to construct precast frames and has a bright prospect.

## Figures and Tables

**Figure 1 materials-15-05791-f001:**
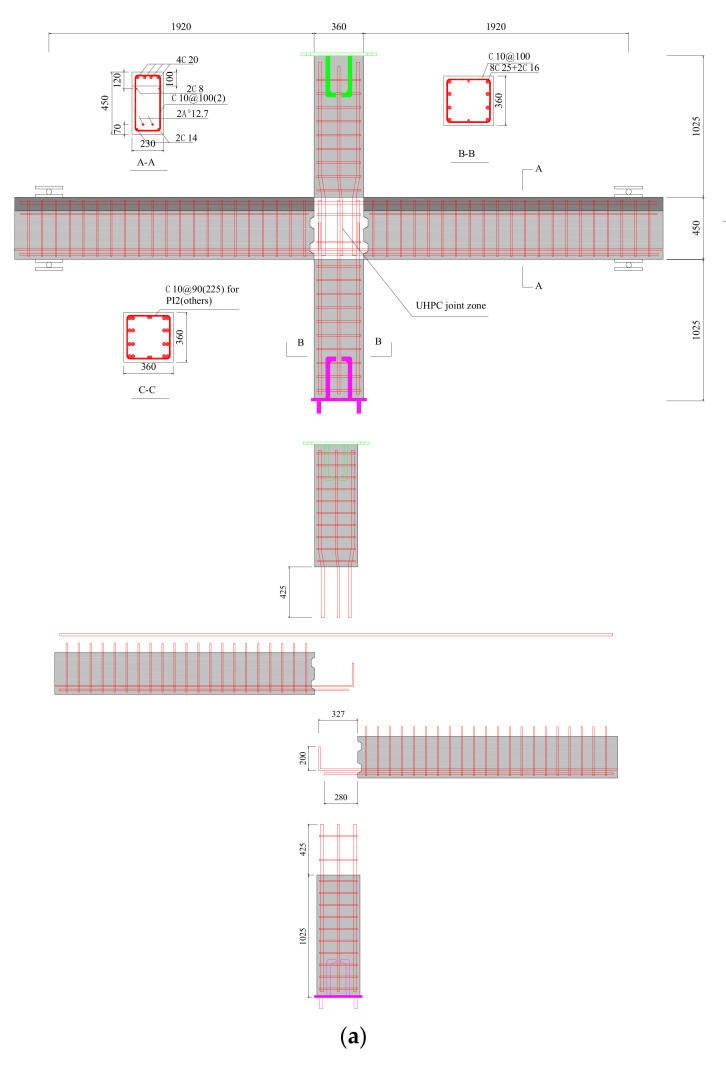
Reinforcement details of specimens. (**a**) UHPC precast specimens, (**b**) monolithic RC specimen.

**Figure 2 materials-15-05791-f002:**
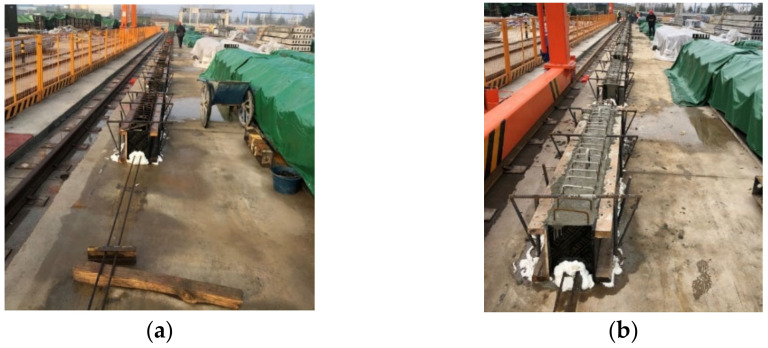
Fabrication process of specimens. (**a**) Pre-tensioning prestressed strands, (**b**) precast beams, (**c**) precast columns, (**d**) assembling of precast specimen, (**e**) UHPC in the joint zone, (**f**) cast-in-place specimen.

**Figure 3 materials-15-05791-f003:**
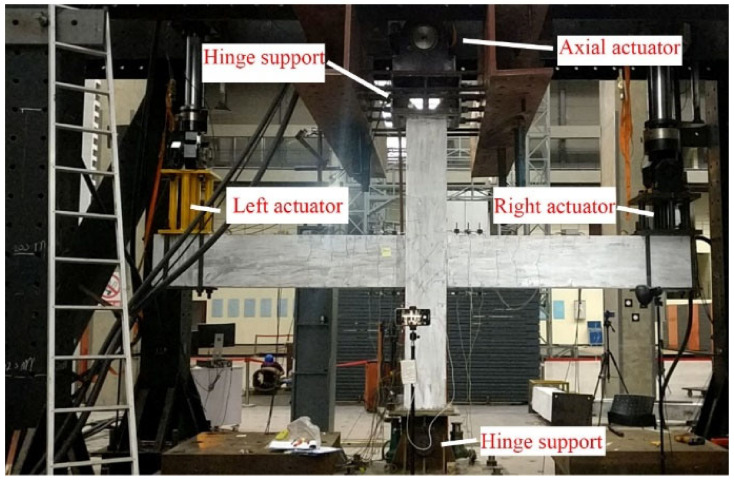
Test setup.

**Figure 4 materials-15-05791-f004:**
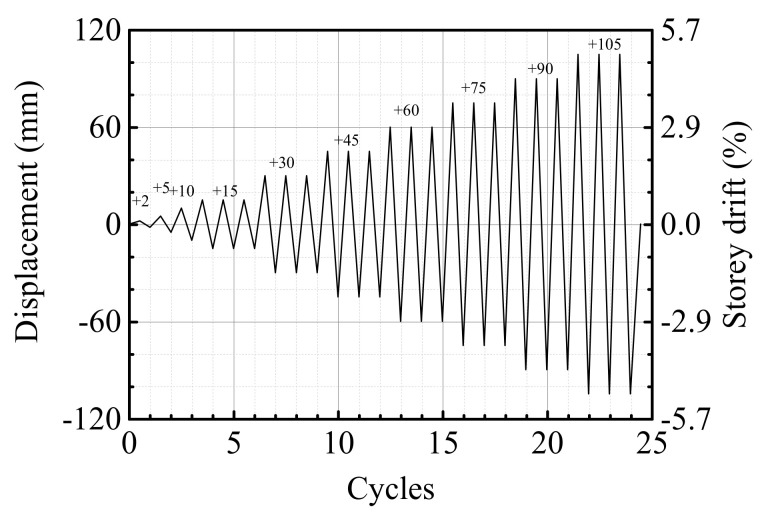
Cyclic loading history.

**Figure 5 materials-15-05791-f005:**
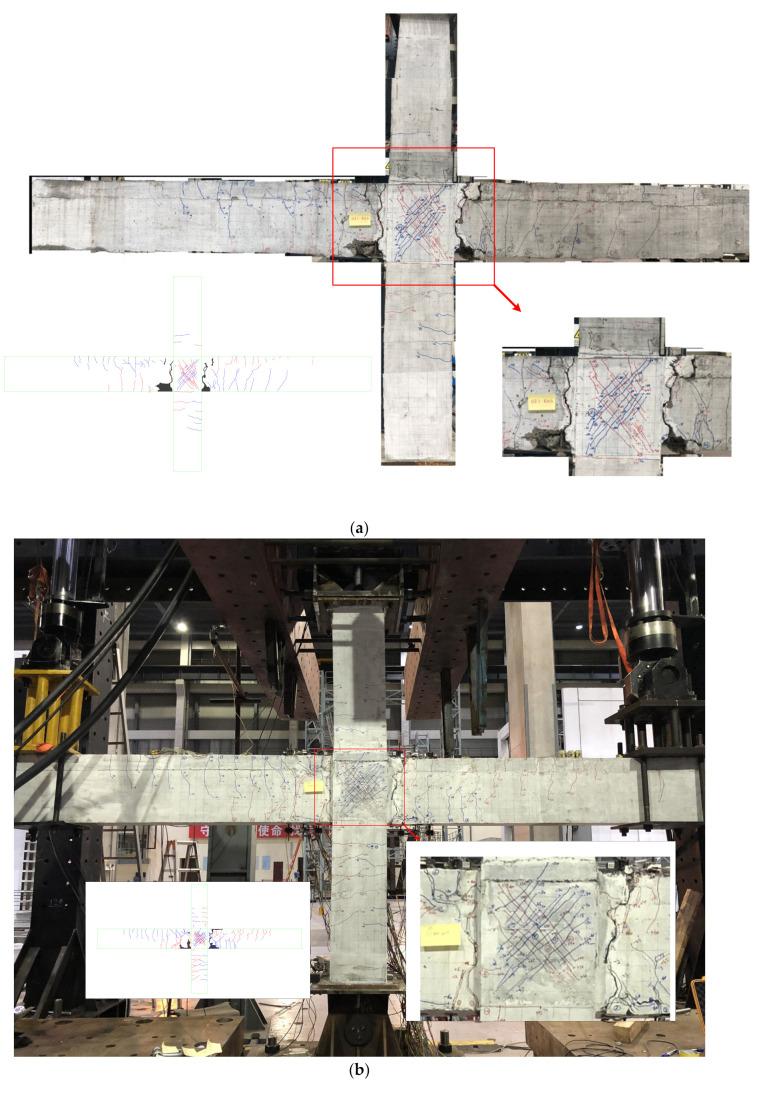
Failure modes. (**a**) Specimen PI1, (**b**) Specimen PI2, (**c**) Specimen PI3, (**d**) Specimen PI4, (**e**) Specimen PI5, (**f**) Specimen PI6, (**g**) Specimen RI1.

**Figure 6 materials-15-05791-f006:**
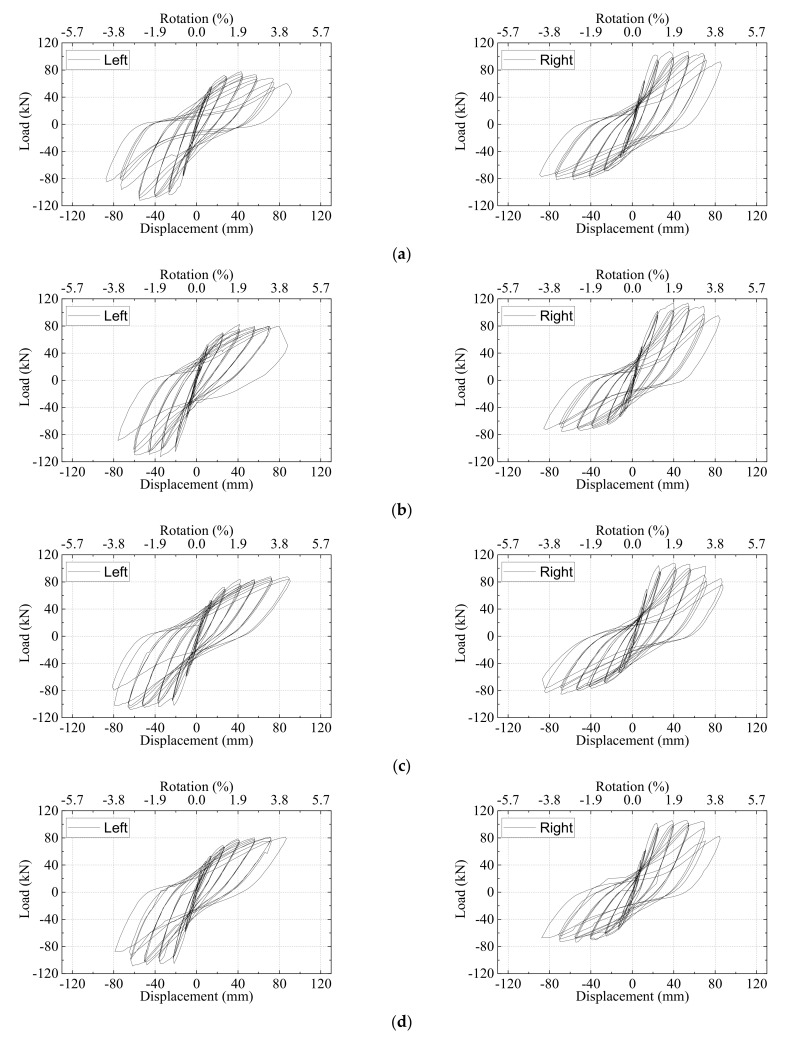
Hysteresis load–displacement curves. (**a**) Specimen PI1-L and PI1-R, (**b**) Specimen PI2-L and PI2-R, (**c**) Specimen PI3-L and PI3-R, (**d**) Specimen PI4-L and 4-R (**e**), Specimen PI5-L and PI5-R (**f**), Specimen PI6-L and PI6-R, (**g**) Specimen RI1-L and RI1-R.

**Figure 7 materials-15-05791-f007:**
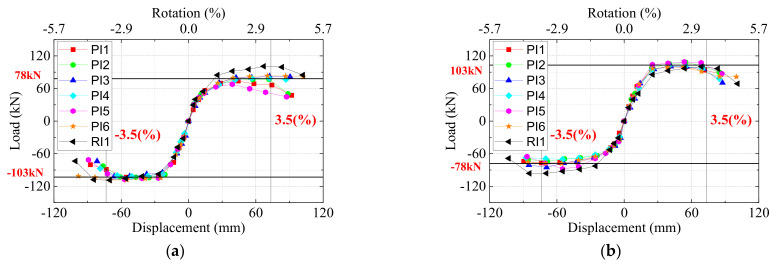
Load–displacement envelope curves. (**a**) Left beams, (**b**) right beams.

**Figure 8 materials-15-05791-f008:**
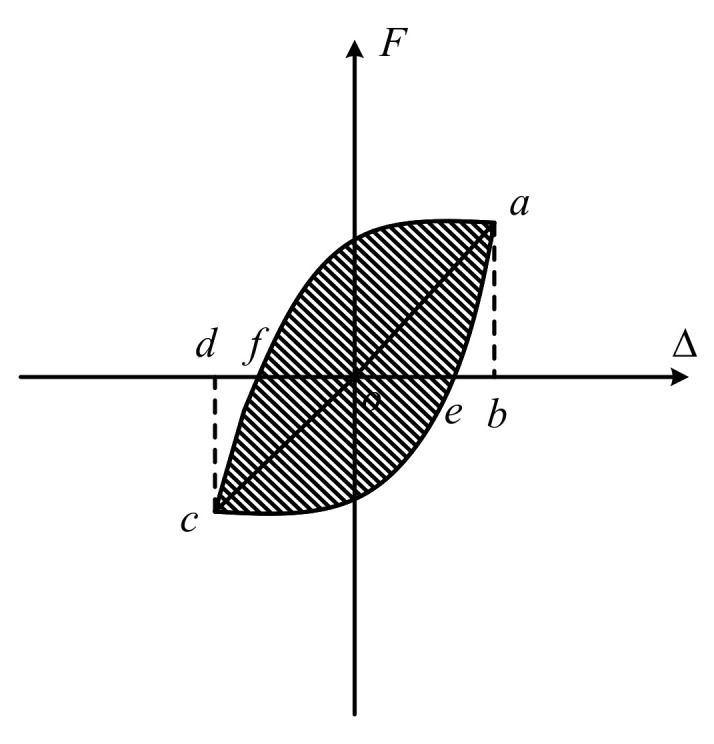
An illustration of hysteresis loop and energy dissipation.

**Figure 9 materials-15-05791-f009:**
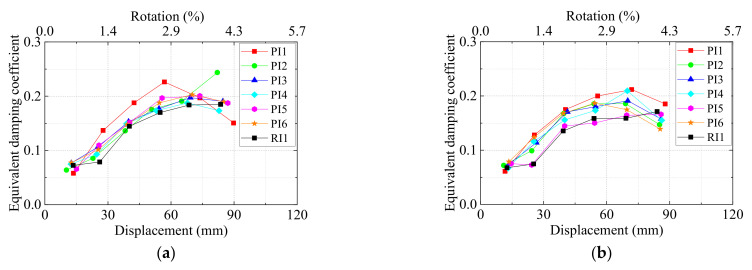
Equivalent damping coefficient. (**a**) Left beams, (**b**) right beams.

**Figure 10 materials-15-05791-f010:**
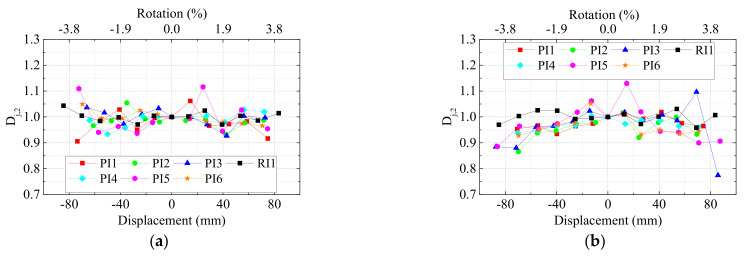
Strength degradation coefficient 
$D_{j,2}$
. (**a**) Left beams and (**b**) right beams.

**Figure 11 materials-15-05791-f011:**
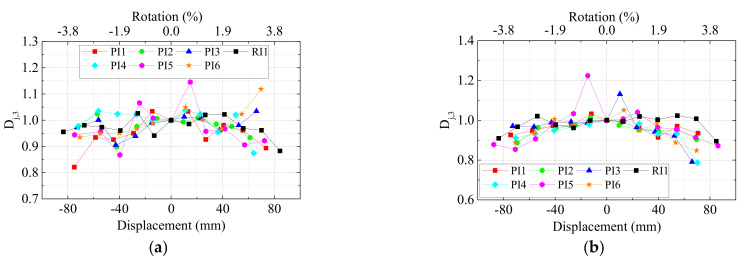
Strength degradation coefficient 
$D_{j,3}$
 of (**a**) left beams and (**b**) right beams.

**Figure 12 materials-15-05791-f012:**
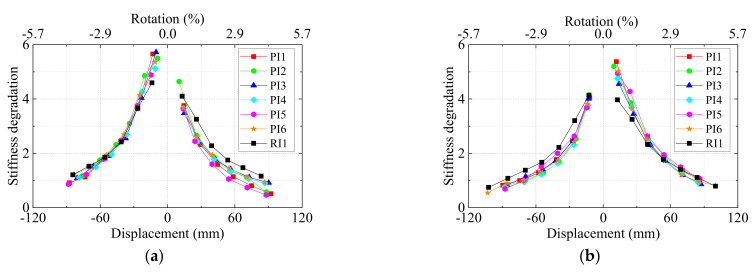
Stiffness degradation. (**a**) Left beams and (**b**) right beams.

**Figure 13 materials-15-05791-f013:**
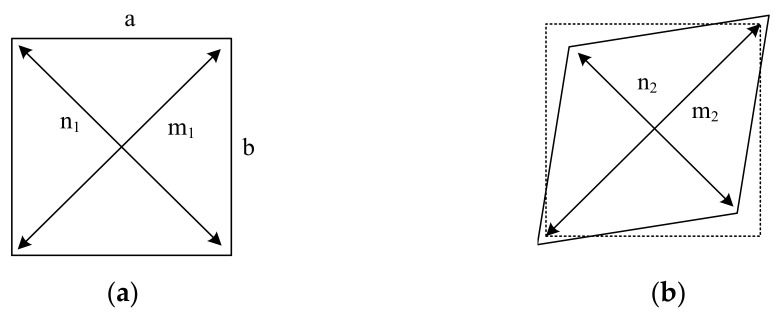
Shear deformation of the joint zone.

**Figure 14 materials-15-05791-f014:**
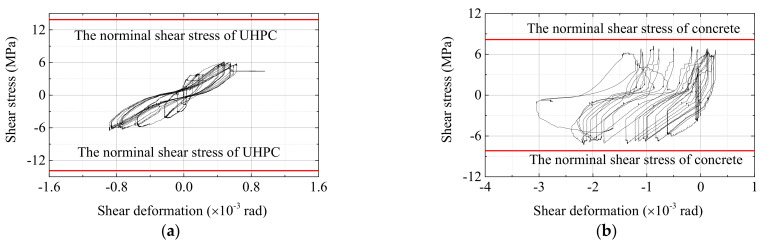
Shear stress versus shear distortion. (**a**) Specimen PI1 and (**b**) Specimen RI1.

**Figure 15 materials-15-05791-f015:**
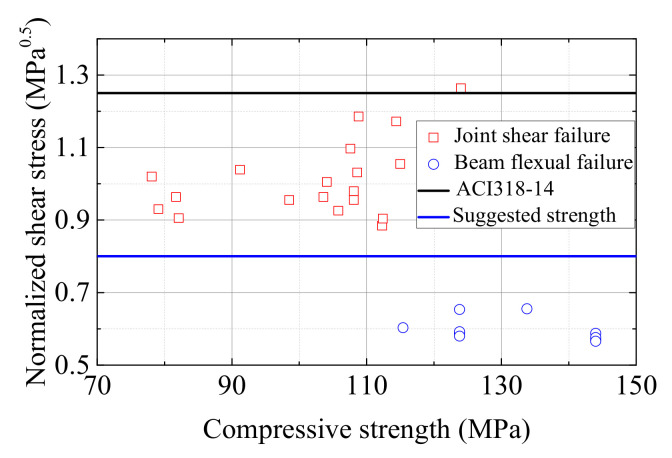
Scatter plot of compressive strength of UHPC versus normalized shear stress.

**Figure 16 materials-15-05791-f016:**
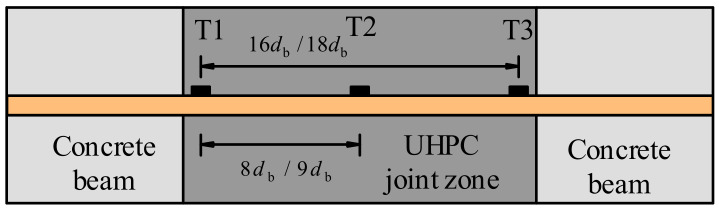
Layout of the strain gauges on the beam top continuous rebars.

**Figure 17 materials-15-05791-f017:**
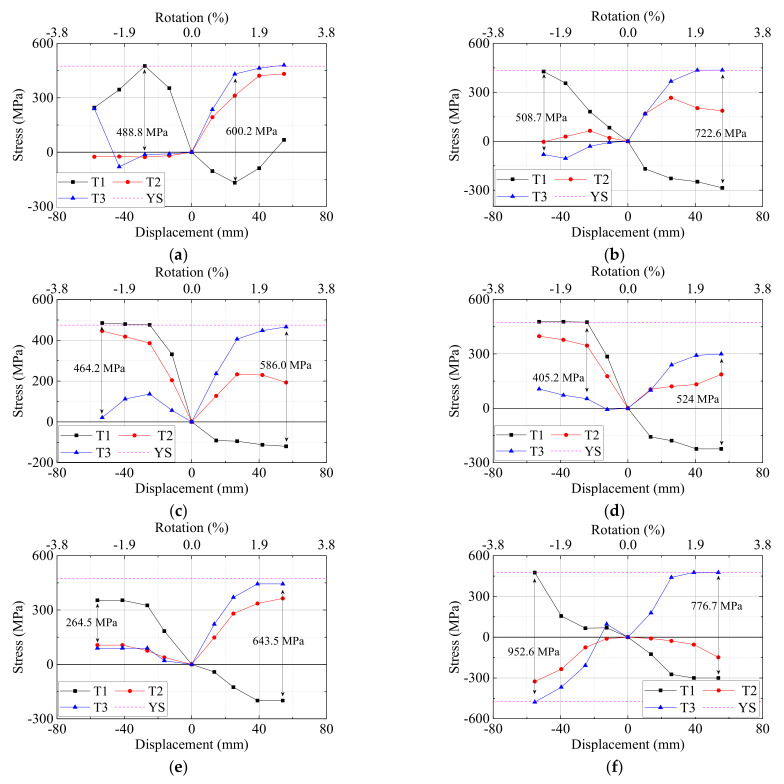
Stress versus displacement envelope curves. (**a**) Specimen PI1, (**b**) Specimen PI2, (**c**) Specimen PI3, (**d**) Specimen PI4, (**e**) Specimen PI5, (**f**) Specimen PI6.

**Table 1 materials-15-05791-t001:** Key parameters of the specimens.

Specimen	Fabrication Method	Beam Top Rebar	Beam Bottom Rebar	Stirrup in Joint	Joint Material	Steel Fiber Volume in UHPC (%)	Axial Force(kN)	$L_{\mathrm{lap}}$ (mm)
PI1	Precast	4C20	2C14+2A^s^12.7	2C10(2)	UHPC100	1.5	600	400
PI2	Precast	5C18	2C14+2A^s^12.7	5C10(2)	UHPC100	1.5	200	400
PI3	Precast	4C20	2C14+2A^s^12.7	2C10(2)	UHPC120	1.5	200	400
PI4	Precast	4C20	2C14+2A^s^12.7	2C10(2)	UHPC100	1.5	200	400
PI5	Precast	4C20	2C14+2A^s^12.7	2C10(2)	UHPC100	1	200	400
PI6	Precast	4C20	2C14+2A^s^12.7	2C10(2)	UHPC100	1.5	200	continuous
RI1	Cast-in place	5C18	4C18	5C10(2)	C50	--	200	--

Notes: 
$L_{\mathrm{lap}}$
 denotes the lap-spliced length of the column rebar.

**Table 2 materials-15-05791-t002:** Mix proportions of UHPC material.

UHPC Type	Cement	Sand	Silica Fume	Water	Superfine Powder	Steel Fiber	Efficiency Water Reducer
UHPC100	1	1.82	0.27	0.29	0.55	0.27	0.011
UHPC120	1	1.53	0.3	0.24	0.3	0.23	0.01

**Table 3 materials-15-05791-t003:** Concrete material properties.

Type of Concrete	Place Used	Cylinder Compressive Strength (MPa)
C50	Beam topping and cast-in-place specimen	42.7
	Precast beam	55.2
	Precast column	45.1
UHPC100 (1%)	Joint zone	115.4
UHPC100 (1.5%)	Joint zone	123.8
UHPC120 (1.5%)	Joint zone	133.8

**Table 4 materials-15-05791-t004:** Mechanical parameters of rebars and strands.

Steel Reinforcement	Diameter (mm)	Yield Strength (MPa)	Ultimate Strength (MPa)
Stirrup	10	513	650
	12	483	692
Longitudinal reinforcement rebar	14	463	671
	16	462	703
	18	436	685
	20	475	681
	25	461	699
Strand	12.7	1581	1860

**Table 5 materials-15-05791-t005:** Summary of test results.

Specimen	Beam	Direction	Yield		Peak		Ultimate		Ductility μ	Peak Shear Stress (MPa)
			$\Delta_{y}$ (mm)	$P_{y}$ (kN)	$\Delta_{m}$ (mm)	$P_{m}$ (kN)	$\Delta_{u}$ (mm)	$P_{u}$ (kN)		
PI1	Left	Positive	21.66	63.01	44.31	73.64	77.78	62.60	3.59	6.54
		Negative	−20.19	−89.49	−40.75	−103.83	−74.34	−88.25	3.68	
	Right	Positive	21.09	87.39	39.40	100.16	86.20	86.46	4.09	
		Negative	−24.76	−65.28	−57.32	−76.23	−89.53	−74.01	3.62	
PI2	Left	Positive	23.66	63.66	42.21	78.08	78.42	66.37	3.31	6.59
		Negative	−20.78	−98.43	−35.07	−104.08	−71.70	−88.47	3.45	
	Right	Positive	25.82	98.61	54.89	108.11	81.97	91.89	3.17	
		Negative	−24.45	−63.56	−68.33	−73.95	−85.59	−70.73	3.50	
PI3	Left	Positive	28.97	71.35	72.23	82.33	90.59	81.29	3.13	7.58
		Negative	−21.92	−90.79	−52.45	−101.56	−74.58	−86.57	3.40	
	Right	Positive	24.56	91.08	41.60	100.61	77.79	85.52	3.17	
		Negative	−32.15	−69.34	−69.16	−85.08	−84.92	−80.93	2.64	
PI4	Left	Positive	26.23	65.79	56.44	78.86	86.87	76.54	3.31	6.45
		Negative	−21.86	−94.43	−50.56	−101.93	−78.80	−87.25	3.61	
	Right	Positive	23.79	91.63	39.74	101.39	78.01	86.18	3.28	
		Negative	−21.34	−58.21	−54.88	−70.25	−87.58	−67.13	4.10	
PI5	Left	Positive	18.61	56.45	39.05	67.74	58.79	57.58	3.16	6.75
		Negative	−23.08	−96.19	−56.99	−107.50	−75.96	−91.37	3.29	
	Right	Positive	24.30	101.40	53.62	108.85	82.18	92.52	3.38	
		Negative	−27.99	−71.03	−54.95	−88.01	−74.00	−74.81	2.64	
PI6	Left	Positive	28.01	71.20	70.36	83.30	102.37	81.43	3.66	6.48
		Negative	−22.99	−98.65	−69.80	−108.54	−98.16	−101.38	4.27	
	Right	Positive	21.70	94.61	38.71	99.45	80.43	84.54	3.71	
		Negative	−25.23	−68.47	−55.68	−77.97	−85.77	−75.85	3.40	
RI1	Left	Positive	30.93	87.44	67.13	100.03	101.99	84.75	3.30	7.27
		Negative	−28.54	−99.81	−70.12	−108.72	−99.13	−93.11	3.47	
	Right	Positive	30.66	88.36	68.98	100.07	90.83	85.06	2.96	
		Negative	−29.95	−84.65	−69.41	−96.08	−94.16	−81.67	3.14	

**Table 6 materials-15-05791-t006:** Summary of bond stresses within UHPC joint area.

Specimen	$\Delta \sigma_{1}(\mathrm{MPa})$	$\Delta \sigma_{2}(\mathrm{MPa})$	s	τ(MPa)	$l/d_{b}$
PI1	510.5	600.2	$8d_{b}$	15.9	9.4
PI2	494.6	722.6	$9d_{b}$	13.7	13.1
PI3	431.5	586	$8d_{b}$	13.5	10.9
PI4	411	524	$8d_{b}$	12.8	10.2
PI5	563.5	643.5	$8d_{b}$	17.6	9.1

**Table 7 materials-15-05791-t007:** Test results for comparison with acceptance criteria in ACI 374.1-05.

Specimen	$$P_{n}/P_{\max}$$		*β*	$$K_{0}/K_{\mathrm{initial}}$$	
	Positive	Negative		Positive	Negative
PI1	0.79	0.81	0.23	0.09	0.06
PI2	0.87	0.88	0.22	0.11	0.06
PI3	0.86	0.92	0.24	0.15	0.06
PI4	0.82	0.87	0.25	0.17	0.06
PI5	0.82	0.75	0.22	0.12	0.10
PI6	0.84	0.86	0.24	0.08	0.12
RI1	0.92	0.91	0.23	0.06	0.07
Acceptance criteria	≥0.75		≥0.125	≥0.05	

Notes: The definitions in the table could be found in ACI374.1-05.

## Data Availability

Not applicable.

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
