# Peer review of "Experimental Study on Seismic Performance of Precast Pretensioned Prestressed Concrete Beam-Column Interior Joints Using UHPC for Connection"

_materials, 2022, doi:10.3390/ma15165791_

Round 1

Reviewer 1 Report

In this paper, the authors reported the investigation on the seismic performance of a novelty joint consisting of precast pretensioned prestressed concrete beam, ordinary precast reinforced concrete (RC) column and UHPC joint zone. Variables such as the axial force, the compressive strength of UHPC, and the stirrup ratio were considered in the tests under low-cyclic loads. The test results indicate the proposed joints exhibit comparable seismic performance of the monolithic RC joint.

The innovation of the research is sufficient, and the results are well analysed and discussed, meeting the requirements for publication. As a no expert on the building structures, I can read the paper and pick up the information about the new solutions in building structures and the test methods of their seismic performance.

Author Response

Response: We appreciate the reviewer’s positive and constructive comment. Since the reviewer did not ask us to revise the paper in specific areas. We appreciate it a lot for the trust and will try our best to revise the paper according to other reviewers’ comments.

Reviewer 2 Report

Experimental study on seismic performance of precast pre-tensioned prestressed concrete beam-column interior joints using UHPC for connection

This study is well presented especially from an engineering point of view with a lot of experimental data. Explanations of experimental data, as well as sketches and schematics, are clearly shown and explained in the manuscript. In my opinion, this manuscript has a minor revision, and regarding this, I suggest to the authors that in the introductory part as well as in the discussion, they should add references relevant to this work and compare their results with the existing ones.

In addition to the engineering approach in this study, the scientific data are short, so my only major objection is to add references that are adequate for the presented study.

The figures are well displayed and explain the given text in detail.

The experimental data are well and thoroughly processed with a sufficient number of samples. The methodology that the Authors used is very well presented. Future controls should be considered by making a large number of samples and testing UHCP concrete samples.

The conclusion part should be supplemented and expanded with the most important results of this study with emphasis on the benefit of these materials for further use in construction. And briefly give a comparison of why this work gave better and improved data in relation to the previously examined UHCP and RC materials similar to these and the components used to obtain them, as well as the contribution and better possibilities of using the obtained materials in this study.

As far as I have checked, the references are selected from relevant journals related to the topic of the work itself. References should also be supplemented, especially in the introductory part, but also in the part where there is a discussion. for such an extensive work, 36 references are too few.

Line 23: Please rephrase the sentence.

Line 44-48: Put some more references regarding high earthquake areas.

Line 58-63: Please add references regarding text in this paragraph.

Line 350: Text regarding Fig.6 should be put under Fig. 6.

Author Response

Response to the Reviewers’ Comments

The authors are grateful to the reviewers for the constructive comments on the paper. We have revised the paper accordingly and responded to the reviewers’ comments point by point below. The revisions are highlighted in red in the revised manuscript.

Reviewer # 2:

Comment # 1: This study is well presented especially from an engineering point of view with a lot of experimental data. Explanations of experimental data, as well as sketches and schematics, are clearly shown and explained in the manuscript. In my opinion, this manuscript has a minor revision, and regarding this, I suggest to the authors that in the introductory part as well as in the discussion, they should add references relevant to this work and compare their results with the existing ones. In addition to the engineering approach in this study, the scientific data are short, so my only major objection is to add references that are adequate for the presented study.

Response: We appreciate the reviewer’s positive and constructive comment. We supplement the contents of other researchers’ studies and more references in the introductory and discussion part about some test results relevant to the work which are highlight in red. In general, in the introductory part, we add the contents of the introduction of other methods of improving the seismic performance of beam column joints, such as FRCC jacketing, CFRP sheets and additional bars. (Line 50 – 61) Also, we add the contents of the importance and novelty of the presented studies. Firstly, the beam section sizes of prestressed concrete members are different from ordinary concrete members. This leads to a greater seismic action to prestressed members. (Line 62 – 69) Secondly, as mentioned in the manuscript, no pretensioned prestressed beams are used in precast concrete frames before, for the difficulty to meet the requirement of the anchorage length of pretensioned strands when ordinary concrete is used in the joint area. Yet with the use of UHPC in the joint zone, we can greatly shorten the anchorage length of pretensioned strands so that it is feasible to combine UHPC joint with pretensioned prestressed members together to improve the structural performances of moment frames. The adequate length of anchorage becomes a key issue to make this kind of structure possible. From the authors’ tests, the length of 40 times of strand diameter is adequate for UHPC joint to build yielding stress in the strands. The detailed discussion in the manuscript indicates that this kind of structure can exhibit comparable seismic performance to monolithic joints. In the red highlight part in introduction and discussion parts, we emphasis the novelty of the present study to the existing studies. (Line 99-103, Line 108-110, Line 114-130, Line 136-139, Line 402-408, Line 442-443, Line 566-572) .

Comment # 2: The figures are well displayed and explain the given text in detail.

The experimental data are well and thoroughly processed with a sufficient number of samples. The methodology that the Authors used is very well presented. Future controls should be considered by making a large number of samples and testing UHCP concrete samples.

 The conclusion part should be supplemented and expanded with the most important results of this study with emphasis on the benefit of these materials for further use in construction. And briefly give a comparison of why this work gave better and improved data in relation to the previously examined UHPC and RC materials similar to these and the components used to obtain them, as well as the contribution and better possibilities of using the obtained materials in this study.

 As far as I have checked, the references are selected from relevant journals related to the topic of the work itself. References should also be supplemented, especially in the introductory part, but also in the part where there is a discussion. for such an extensive work, 36 references are too few.

Response: In the conclusion part, we have revised the manuscript with the highlight sentences by emphasizing the novelty of our studies. The suggested anchorage length of pretensioned strands is novelty. Also, the lap-spliced connection of column rebars in the joint zone rather than grout sleeves connection is a breakthrough achievement in the study. Besides, a shear strength of UHPC joint are studied and suggested as 0.8 , which is a novelty result. And 18 references from the latest studies are added. 

Line 23: Please rephrase the sentence.

Line 44-48: Put some more references regarding high earthquake areas.

Line 58-63: Please add references regarding text in this paragraph.

Line 350: Text regarding Fig.6 should be put under Fig. 6.

Response: Those four flaws are revised accordingly which are highlight in red sentences. (Line 23, Line 44-48, Line 62-69, Line 392)

Reviewer 3 Report

The traditional connections and reinforcement details of precast RC frames are complex and cause difficulty in construction. Ultra-high performance concrete (UHPC) exhibits outstanding compressive strength and bond strength with rebars and strands, thus the usage of UHPC in the joint core area will reduce the amount of transverse reinforcement and shorten the anchoring length of beam rebars as well as strands significantly.

In this paper a novelty joint consisting of precast pretensioned prestressed concrete beam was presented, ordinary precast reinforced concrete (RC) column and UHPC joint zone. Totally six novelty interior joints and one monolithic RC joint were tested under low-cyclic loads to study the seismic performance of the proposed joints.

This is an extensive research, with a lot of experimental analysis. Variables such as the axial force, the compressive strength of UHPC, the stirrup ratio were considered in the tests.

Thematically the work is interesting for the researchers and professionals and the proposed manuscript is relevant to the scope of the journal.

I found it appropriate for publication in the Materials journal, but only after some modifications and clarification from the Authors.

The title is a clear representation of the manuscript's content.

The abstract reflects realistically the substance of the work. 

The overall organization and structure of the manuscript are appropriate. The paper is well written and the topic is appropriate for the journal.
The aim of the paper is well described and the discussion was well approached, its results and discussion are correlated to the cited literature data.

The literature review is comprehensive and properly done. However, it would be even more interesting if a few newer references could be cited in the Introduction section.

The novelty of the work must be more clearly demonstrated. Please add a few sentences which would explain the novelty of the presented results compared to other studies.

The significance of the Work: Given the large number of analyzed data, this is an interesting study with a possible significant impact in this area.

Statistical interpretation of the analytical data must be more properly presented. At least an ANOVA test to show the influence of variables would be interesting addition to discussion section.

Other Specific Comments: The work is properly presented in terms of the language. The work presented here is very interesting and well done, it is presented in a compact manner.
In general, there are no doubtful or controversial arguments in the manuscript. The methodology applied in the research is presented in clear manner, so that it is repeatable by other authors.

The main drawback of the paper could be the extent of novelty (and perhaps an older list of references), or the main novelty in the present work, compared to the works of other researchers? In my opinion, the authors should put additional effort to demonstrate that the present work gives a substantial contribution in the research area.

Author Response

Response to the Reviewers’ Comments

The authors are grateful to the reviewers for the constructive comments on the paper. We have revised the paper accordingly and responded to the reviewers’ comments point by point below. The revisions are highlighted in red in the revised manuscript.

Reviewer # 3:

Comment # 1: The traditional connections and reinforcement details of precast RC frames are complex and cause difficulty in construction. Ultra-high performance concrete (UHPC) exhibits outstanding compressive strength and bond strength with rebars and strands, thus the usage of UHPC in the joint core area will reduce the amount of transverse reinforcement and shorten the anchoring length of beam rebars as well as strands significantly.

In this paper a novelty joint consisting of precast pretensioned prestressed concrete beam was presented, ordinary precast reinforced concrete (RC) column and UHPC joint zone. Totally six novelty interior joints and one monolithic RC joint were tested under low-cyclic loads to study the seismic performance of the proposed joints.

This is an extensive research, with a lot of experimental analysis. Variables such as the axial force, the compressive strength of UHPC, the stirrup ratio were considered in the tests.

Thematically the work is interesting for the researchers and professionals and the proposed manuscript is relevant to the scope of the journal.

I found it appropriate for publication in the Materials journal, but only after some modifications and clarification from the Authors.

The title is a clear representation of the manuscript's content. The abstract reflects realistically the substance of the work.

The overall organization and structure of the manuscript are appropriate. The paper is well written and the topic is appropriate for the journal.

The aim of the paper is well described and the discussion was well approached, its results and discussion are correlated to the cited literature data.

The significance of the Work: Given the large number of analyzed data, this is an interesting study with a possible significant impact in this area.

Statistical interpretation of the analytical data must be more properly presented. At least an ANOVA test to show the influence of variables would be interesting addition to discussion section.

Response: We appreciate the reviewer’s positive and constructive comment. We have checked the paper that using an ANOVA test to show the influence of variables [1]. In the paper, parameters such as aggregate, fiber, error are analyzed to show the influence of compressive strength, splitting tensile strength, and flexural tensile strength of the novel concrete material. The number of concrete specimens is 12 and the degree of freedom is enough. Yet in the authors’ study, the total number of specimen is 6 which is not adequate for a meaningful statistical analysis. Also, the influence of the strength of specimens in the tests are complex. The flexural strength of the beam section also has a great impact on the seismic performance of beam column joints. Herein, it is a routine to not carry on the ANOVA test in the study of seismic performance of beam column joints, which is shown in the literature [2-4].

Nevertheless, we make a great effort on the supplement in the introduction, discussion

and conclusion parts to emphasis the novelty of our study. Please check on the highlight sentences in red to examine the revised work we have done.

[1]Cakir, Ozgur, Ipek, et al. Properties of polypropylene fiber reinforced concrete using recycled aggregates[J]. Construction and Building Materials, 2015.

[2] Shc A , Jhk A , Hj B , et al. Seismic behavior of beam-column joints with different concrete compressive strengths[J]. Journal of Building Engineering, 52.

[3] Deng Z ,  Xu C ,  Zeng J , et al. Seismic performance and shear bearing-capacity of truss SRC beam-column frame joints:[J]. Advances in Structural Engineering, 2021, 24(7):1311-1325.

[4]Zhang J, Pei Z, Rong X. Experimental seismic study of an innovative precast steel–concrete composite beam–column joint[J]. Soil Dynamics and Earthquake Engineering, 2022, 161: 107420.

Comment # 2: Other Specific Comments: The work is properly presented in terms of the language. The work presented here is very interesting and well done, it is presented in a compact manner.

In general, there are no doubtful or controversial arguments in the manuscript. The methodology applied in the research is presented in clear manner, so that it is repeatable by other authors.

The main drawback of the paper could be the extent of novelty (and perhaps an older list of references), or the main novelty in the present work, compared to the works of other researchers? In my opinion, the authors should put additional effort to demonstrate that the present work gives a substantial contribution in the research area.

Response: We have made the supplement in the introduction, discussion

and conclusion parts to compare our studies with the existing ones and emphasis the novelty of our study. Please check on the highlight sentences in red to examine the revised work we have done. We supplement the contents of other researchers’ studies and more references in the introductory and discussion part about some test results relevant to the work which are highlight in red. In general, in the introductory part, we add the contents of the introduction of other methods of improving the seismic performance of beam column joints, such as FRCC jacketing, CFRP sheets and additional bars. (Line 50 – 61) Also, we add the contents of the importance and novelty of the presented studies. Firstly, the beam section sizes of prestressed concrete members are different from ordinary concrete members. This leads to a greater seismic action to prestressed members. (Line 62 – 69) Secondly, as mentioned in the manuscript, no pretensioned prestressed beams are used in precast concrete frames before, for the difficulty to meet the requirement of the anchorage length of pretensioned strands when ordinary concrete is used in the joint area. Yet with the use of UHPC in the joint zone, we can greatly shorten the anchorage length of pretensioned strands so that it is feasible to combine UHPC joint with pretensioned prestressed members together to improve the structural performances of moment frames. The adequate length of anchorage becomes a key issue to make this kind of structure possible. From the authors’ tests, the length of 40 times of strand diameter is adequate for UHPC joint to build yielding stress in the strands. The detailed discussion in the manuscript indicates that this kind of structure can exhibit comparable seismic performance to monolithic joints. In the red highlight part in introduction and discussion parts, we emphasis the novelty of the present study to the existing studies. (Line 99-103, Line 108-110, Line 114-130, Line 136-139, Line 402-408, Line 442-443, Line 566-572) .

In the conclusion part, we have revised the manuscript with the highlight sentences by emphasizing the novelty of our studies. The suggested anchorage length of pretensioned strands is novelty. Also, the lap-spliced connection of column rebars in the joint zone rather than grout sleeves connection is a breakthrough achievement in the study. Besides, a shear strength of UHPC joint are studied and suggested as 0.8  , which is a novelty result.

In all 18 additional references from newest work have been added and discussed in the revised manuscript.

Reviewer 4 Report

The submitted paper materials-1849559 entitled: “Experimental study on seismic performance of precast pretensioned prestressed concrete beam-column interior joints using UHPC for connection is an experimental study that presents a novel technique for reinforcing internal rc beam column joints consisting of precast pretensioned prestressed concrete beam, ordinary precast reinforced concrete (RC) column and UHPC joint zone. A total of six precast interior beam-column joints and one monolithic joint were tested on cyclic loads to study the seismic performance. The paper discusses an interesting field. The experimental program is well planned and the paper has a good structure. The following minor suggestions are raised for Authors’ reference:

  1. The literature review should include more information about the benefits of using prestressed RC joints instead of traditionally reinforced, or reinforced using novel materials and techniques. To help the authors I propose to read and discuss the following recent studies which will also help find others in this direction in the literature.

       Effectiveness of the Novel Rehabilitation Method of Seismically Damaged RC Joints Using C-FRP Ropes and Comparison with Widely Applied Method Using C-FRP Sheets—Experimental Investigation. Sustainability 2021, 13, 6454.

       Behavior of RC Beam–Column Joints Strengthened with Modified Reinforcement Techniques. Sustainability 2022, 14, 1918.

  1. A very important aspect of nowadays structures is to also meet sustainability standards. Is this technique considered sustainable?
  2. Figures 5 a-g must be enlarged or better zoomed in in the joint area so that the comparison would be more obvious, the differences are  barely visible and it is difficult for the reader to follow.
  3. The description of figure 6 must be placed bellow the figure and not aside.

Author Response

Response to the Reviewers’ Comments

The authors are grateful to the reviewers for the constructive comments on the paper. We have revised the paper accordingly and responded to the reviewers’ comments point by point below. The revisions are highlighted in red in the revised manuscript.

Reviewer # 4:

Comment # 1: The submitted paper materials-1849559 entitled: “Experimental study on seismic performance of precast pretensioned prestressed concrete beam-column interior joints using UHPC for connection” is an experimental study that presents a novel technique for reinforcing internal rc beam column joints consisting of precast pretensioned prestressed concrete beam, ordinary precast reinforced concrete (RC) column and UHPC joint zone. A total of six precast interior beam-column joints and one monolithic joint were tested on cyclic loads to study the seismic performance. The paper discusses an interesting field. The experimental program is well planned and the paper has a good structure. The following minor suggestions are raised for Authors’ reference:

  1. The literature review should include more information about the benefits of using prestressed RC joints instead of traditionally reinforced, or reinforced using novel materials and techniques. To help the authors I propose to read and discuss the following recent studies which will also help find others in this direction in the literature.

−       Effectiveness of the Novel Rehabilitation Method of Seismically Damaged RC Joints Using C-FRP Ropes and Comparison with Widely Applied Method Using C-FRP Sheets—Experimental Investigation. Sustainability 2021, 13, 6454.

−       Behavior of RC Beam–Column Joints Strengthened with Modified Reinforcement Techniques. Sustainability 2022, 14, 1918.

Response: We appreciate the reviewer’s positive and constructive comment. Both of the references as well as other relevant ones (in total 18 new references) are added in the revised manuscript.

Comment # 2:2.  A very important aspect of nowadays structures is to also meet sustainability standards. Is this technique considered sustainable?

Response: UHPC is known for better performance on sustainability compared to ordinary concrete for the contribution of steel fiber composites. Also, prestressed members perform better than non-prestressed members on sustainability for the prestressed forces would control better in crack behavior. All the specimens are well designed with Chinese concrete structural codes, which means they have meet the sustainability standards even without the help of UHPC and prestressed forces. The technique proposed in the paper have a better sustainable performance for sure with the contribution of UHPC and prestressed forces.

Comment # 3 and 4 : 3.    Figures 5 a-g must be enlarged or better zoomed in in the joint area so that the comparison would be more obvious, the differences are  barely visible and it is difficult for the reader to follow.

  1. The description of figure 6 must be placed bellow the figure and not aside.

Response: Figure 5a-g have been resized and zoomed in the revised manuscript to be visible for readers. (Line 322). And the position of description of figure 6 has been corrected. (Line392)
